# Clinical Role of Upfront F-18 FDG PET/CT in Determining Biopsy Sites for Lung Cancer Diagnosis

**DOI:** 10.3390/diagnostics14020153

**Published:** 2024-01-09

**Authors:** Byunggeon Park, Jae-Kwang Lim, Kyung Min Shin, Jihoon Hong, Jung Guen Cha, Seung Hyun Cho, Seo Young Park, Hun Kyu Ryeom, See Hyung Kim, An Na Seo, Seung-Ick Cha, Jaehee Lee, Hoseok Lee, Jongmin Park

**Affiliations:** 1Department of Radiology, School of Medicine, Kyungpook National University, Daegu 41944, Republic of Korea; 2Department of Pathology, School of Medicine, Kyungpook National University, Daegu 41944, Republic of Korea; 3Department of Internal Medicine, School of Medicine, Kyungpook National University, Daegu 41944, Republic of Korea; 4Department of Radiology, Semyung Radiology Clinic, Gumi 39254, Gyeongsangbuk-do, Republic of Korea

**Keywords:** lung cancer, biopsy, positron emission tomography computed tomography

## Abstract

Purpose: This study aimed to investigate the impact of FDG PET/CT timing for biopsy site selection in patients with stage IV lung cancer regarding complications and diagnostic yield. Methods: This retrospective analysis was performed on 1297 patients (924 men and 373 women with a mean age of 71.4 ± 10.2 years) who underwent percutaneous needle biopsy (PNB) for stage IV lung cancer diagnosis in two hospitals. Data collected included the patient’s characteristics, order date of the biopsy and PET/CT exams, biopsy target site (lung or non-lung), guidance modality, complications, sample adequacy, and diagnostic success. Based on the order date of the PNB and PET/CT exams, patients were categorized into upfront and delayed PET/CT groups. Results: PNB for non-lung targets resulted in significantly lower rates of minor (8.1% vs. 16.2%), major (0.2% vs. 3.4%), and overall complications (8.3% vs. 19.6%) compared to PNB for lung targets (*p* < 0.001 for all types of complications). Compared to the delayed PET/CT group, the upfront PET/CT group exhibited a lower probability of lung target selection of PNB (53.9% vs. 67.1%, *p* < 0.001), including a reduced incidence of major complications (1.0% vs. 2.9%, *p* = 0.031). Moreover, there was no significant difference in the occurrence of minor and total complications between the two groups. Upfront PET/CT and delayed PET/CT groups showed no significant difference regarding sample adequacy and diagnostic success. Conclusions: Upfront PET/CT may have an impact on the selection of the biopsy site for patients with advanced lung cancer, which could result in a lower rate of major complications with no change in the diagnostic yield. Upfront PET/CT demonstrates potential clinical implications for enhancing the safety of lung cancer diagnosis in clinical practice.

## 1. Introduction

Over the past few decades, image-guided percutaneous needle biopsy (PNB) for the lung has emerged as the preferred procedure for lung cancer diagnosis [1,2]. Compared to transbronchial or surgical biopsy, PNB under CT, US, and fluoroscopy guidance can be performed easily under local anesthesia, without the need for general anesthesia [3,4,5,6]. PNB for the lung has limitations due to the potential risk of pneumothorax in approximately 15–38% of patients, with a subsequent requirement for chest tube insertion in approximately 5–10% of patients [3,5,7]. To avoid the risks of pneumothorax and chest tube placement, a histopathological diagnosis has been obtained through PNB for metastasis involving the pleura, lymph nodes, bones, and solid organs other than the lung [4,8,9,10,11,12]. Detection of tumor viability through PET/CT improved the diagnostic yield of PNB for lung cancer diagnosis in patients who underwent a biopsy of the lung or other structures [12,13,14,15,16]. Prior studies that compared PNB with and without F-18 FDG PET/CT revealed that utilizing F-18 FDG PET/CT for guiding such biopsies enhanced the accuracy of diagnosis [16,17]. PNB guided by PET/CT findings exhibited a remarkable diagnostic success rate of 98.1% for lymph nodes and 96.1% for bones in the diagnosis of lung cancer [12,14].

The use of fluorine-18 fluorodeoxyglucose (F-18 FDG) PET/CT has been recommended by several guidelines for lung cancer assessment [18,19,20,21]. According to clinical guidelines by the National Comprehensive Cancer Network, the British Thoracic Society, and Cancer Council Australia, the utilization of F-18 FDG PET/CT before additional diagnostic procedures is recommended [20,22,23,24]. It has the capability to detect metastases, potentially offering an alternative means of obtaining tissue for pathological diagnosis. F-18 FDG-PET/CT serves a dual role by guiding the biopsy procedure and contributing to the assessment of the stage of the disease. Performing F-18 PET/CT before biopsy in patients with lung cancer could potentially reduce the need for additional invasive procedures [20,24]. When planning a diagnostic strategy for lung cancer, the diagnostic yield and potential risks involved should be carefully considered [20].

The relationship between the timing of F-18 FDG PET/CT in the diagnosis of lung cancer and its impact on the safety and diagnostic yield has rarely been reported in the literature. Therefore, this study aimed to compare the safety and diagnostic yield of PNB between lung and non-lung lesions for lung cancer diagnosis and evaluate the impact of PET/CT imaging prior to PNB on safety and diagnostic yield in patients with advanced lung cancer.

## 2. Materials and Methods

The institutional review board of the authors’ institutions approved this retrospective study and waived the requirement for informed consent.

### 2.1. Study Population

Patients with pathologically confirmed lung cancer at Kyungpook National University Hospital and Kyungpook National University Chilgok Hospital, Daegu, Republic of Korea from January 2012 to December 2021 were retrospectively reviewed. The inclusion criteria included (a) patients with newly diagnosed stage IV lung cancer who had not received any treatment before and (b) patients who underwent only one biopsy.

A flowchart of patient inclusion is shown in Figure 1. A total of 464 patients were excluded from this study due to the following: (a) inappropriate pathology results (*n* = 9), (b) other primary malignancies at the time of examination (*n* = 89), (c) unknown biopsy site (*n* = 33), (d) fine-needle aspiration (*n* = 119), (e) histologic examination other than PNB (*n* = 193), and (f) patients who underwent chest CT, PET/CT study and biopsy for 4 weeks (*n* = 21). Finally, a total of 1297 patients who underwent PNB (mean age of 71.4 ± 10.2 years), consisting of 924 men and 373 women, were subjected to analysis. The biopsy sites were classified into lung target and non-lung target, including the pleura, lymph nodes, solid organs, bones, and soft tissues. Patients were classified into two groups: the upfront PET/CT group, comprised of patients where the biopsy site was determined after performing PET/CT, and the delayed PET/CT group, comprised of patients where the biopsy site was determined prior to performing PET/CT.

### 2.2. Chest CT and F-18 FDG PET/CT Image Acquisition

All patients underwent chest CT encompassing the jaw to the upper abdomen area using various multidetector CT scanners, including 16-, 64-, and 128-slice scanners. CT examinations were obtained in the supine position and full inspiration with or without contrast media. All CT scans were reconstructed using a sharp or standard 1.0–3.0-mm slice thickness reconstruction kernel with a 100- or 120-kVp tube voltage. An automatic tube current was used for all patients. All CT images were reviewed by four thoracic radiologists.

The timing of PET/CT examinations is solely determined based on the patient’s schedule. Before the administration of F-18 FDG, all patients fasted for at least 6 h, and their blood glucose levels were assessed. Patients with blood glucose levels higher than 150 mg/dL were rescheduled. Intravenous injections of approximately 3.7–5.55 MBq of FDG per kilogram of body weight were administered to the patients; they were asked to rest for 1 h before image acquisition. The 18F-FDG PET/CT scans were performed using Discovery 600 or 690 (GE Healthcare, Chicago, IL, USA). Before the PET scan, a low-dose CT scan without contrast media was obtained from the skull vertex to the knees in the supine position under quiet breathing. PET scans with maximum spatial resolutions of 5.1 mm (Discovery 600) and 4.9 mm (Discovery 690) were also performed from the skull vertex to the knees at 1.5 min per bed position. PET images were reconstructed with a 192 × 192 matrix, an ordered-subset expectation maximum iterative reconstruction algorithm, with a slice thickness of 3.27 mm. All F-18 FDG PET/CT images were reviewed by two experienced nuclear medicine physicians.

### 2.3. PNB

The determination of the biopsy site was based on the consensus among experienced specialists in the departments of radiology, pulmonary medicine, nuclear medicine, and thoracic and cardiovascular surgery, using all available imaging studies, including old film, MRI, and bone scan. All procedures were performed by two experienced radiologists (with 20 years and 15 years of experience in image-guided PNB, respectively). Representative cases are shown in Figure 2, Figure 3 and Figure 4. Immediately after the procedure, CT or US was performed to evaluate for acute complications such as active bleeding, pneumothorax, hemothorax, hematoma, and parenchymal hemorrhage. Thereafter, a chest radiograph was performed to determine the post-procedural pneumothorax or parenchymal hemorrhage within 3 h and 24 h for patients who had undergone lung or pleural biopsy. Also, all patients were monitored for signs and symptoms secondary to post-procedure-related complications. The number of punctures depended on the specimen quality and the patient’s tolerance. All specimens were immediately fixed in 10% formalin and sent for histopathological examination.

### 2.4. CT-Guided PNB

Procedures were performed under CT guidance on a 16-section multidetector CT (LightSpeed 16; GE Healthcare, Milwaukee, WI, USA) or 128-section multidetector CT (Revolution Evo; GE Healthcare, Milwaukee, WI, USA) with an 18-gauge core biopsy needle (ProMag; Argon Medical Devices Inc., Athens, TX, USA and Bard Magnum; Bard, Covington, GA, USA) without a coaxial approach for lung target and Bonopty 12-gauge coaxial biopsy system (Apriomed; Londonbery, NH, USA) for bone target.

### 2.5. C-Arm Cone-Beam CT-Guided PNB

All biopsies were performed using one of two cone-beam CT systems (Azurion; Philips Healthcare, Best, Netherlands, and Innova 4100-IQ; GE Healthcare, Chicago, IL, USA) with or without the aid of a virtual guidance software program (XperGuide version 1.65; Philips Healthcare, Bothell, WA, USA). To identify the lung target, pre-procedural cone-beam CT imaging was performed. When virtual guidance was used, the vertical alignment from the skin entry site to the target lesion was automatically established. The needle was then inserted under fluoroscopy guidance. All biopsies were performed using an 18-gauge biopsy needle (Bard Magnum; Bard, Covington, GA, USA).

### 2.6. US-Guided PNB

US-guided PNB was performed using two US systems (EPIQ Elite; Philips Healthcare, Bothell, WA, USA, and iU22; Philips Healthcare, Bothell, WA, USA). Low-frequency (1–5 MHz) or high-frequency (10–12 MHz) transducers were selected according to the patient’s body habitus to ensure ample visualization of the lesion. All samples were acquired using an 18-gauge automated core biopsy needle (Acecut; TSK Laboratory, Tochigi, Japan) without a coaxial approach.

### 2.7. Pathological Analysis

All pathological specimens were formalin-fixed, paraffin-embedded, and stained with hematoxylin and eosin with immunohistochemical staining. All histopathological specimens were evaluated by two pathologists, with more than 10 years of experience, according to the fourth edition of the World Health Organization. The histopathological diagnoses from initial PNB were classified into three categories: malignant, nonspecific benign including atypical cells, and nondiagnostic (e.g., insufficient specimens and fibromuscular tissue).

### 2.8. Diagnostic Assessment and Complication Analysis

The reference standard for each lesion was established in one of three ways through medical record and follow-up image review [25]: If the biopsy result revealed malignant cells and the clinical course was consistent with obvious malignant processes, the lesion was confirmed to be a malignancy if the target lesion was surgically resected, the surgical pathologic report was considered as the final diagnosis, and lesions that did not fulfill the criteria were classified as having incomplete reference standards and were excluded.

If the result of the samples obtained from PNB was nondiagnostic, the samples were considered inadequate. Otherwise, the samples were considered sufficient. Malignancy established through a pathologic review of samples obtained by PNB was deemed a diagnostic success. Diagnostic failure was defined as a nonspecific benign or nondiagnostic result.

Post-procedural complications were assessed during the hospitalization period and categorized as minor (class A–B) or major (class C–F) according to the Society of Interventional Radiology classification [26]. Procedure time was defined as the time interval between the first and last image. Hospital stay data were acquired via a medical record review.

### 2.9. Statistical Analysis

All statistical analyses were performed using R software (R version 4.0.3, The R Foundation for Statistical Computing). Continuous variables were tested for normality using the Shapiro–Wilk test. The independent samples *t*-test was used for normally distributed data expressed as means and standard deviation. Categorical variables are expressed as frequencies with percentages. The x^2^ test was performed to test for proportional differences. Statistical significance was set to *p* < 0.05.

## 3. Results

### 3.1. Patient Characteristics

The patient characteristics are summarized in Table 1. The patients were divided into two groups: the upfront PET/CT group (39.3%, 510/1297) and the delayed PET/CT group (60.7%, 787/1297) based on PET/CT timing. A coaxial system was used in 2.8% (36/1297) of PNB for bone targets and a non-coaxial technique in 97.2% (1261/1297). The most frequent location for the target biopsy was the lung (61.9%, 803/1297), followed by the pleura (15.8%, 205/1297), lymph nodes (10.3%, 133/1297), bone (6.8%, 88/1297), liver (3.3%, 43/1297), soft tissues (1.8%, 23/1297), and kidneys (0.2%, 2/1297).

The diagnoses included non-small cell lung cancer not otherwise specified (14.0%, 181/1297), adenocarcinoma (53.4%, 692/1297), squamous cell carcinoma (18.3%, 237/1297), large cell carcinoma (1.0%, 13/1297), sarcomatoid carcinoma (0.8%, 10/1297), and small cell lung cancer (12.6%, 164/1297).

No statistically significant differences were observed in age, sex, biopsy technique, and histological diagnosis between the upfront and delayed PET/CT groups.

### 3.2. Biopsy Characteristics and Outcomes

Table 2 shows that PNB was performed in 494 patients in a non-lung target and 803 patients in the lung target. There was a significantly higher number of cores obtained from the non-lung target compared to the lung target (2.4 ± 1.0 vs. 1.6 ± 0.8; *p* < 0.001). A significantly shorter PNB procedure time was noted in the non-lung target compared to the lung target (643.7 ± 309.6 vs. 781.6 ± 374.3; *p* < 0.001). Guidance modality during PNB was significantly different (*p* < 0.001). The biopsy site selection for the lung target (56.2%; 451/803 vs. 43.8%; 352/803) and non-lung target (45.1%; 223/494 vs. 54.9%; 271/494) significantly differed between radiologist A and radiologist B (*p* < 0.001). There was no difference in sex, age, and hospital stay between PNB for the lung and non-lung targets.

The overall sample adequacy and diagnostic success rate were 97.5% and 93.4%, respectively. In the subgroup analysis, according to the biopsy site between the lung target and non-lung target, sample adequacy (97.8% vs. 97.2%, *p* = 0.629) and diagnostic success rate (93.4% vs. 93.3%, *p* = 1.000) were not significantly different; thus, an analysis was conducted regarding factors affecting sample adequacy and diagnostic success (Appendix A). No significant differences were observed in the number of cores obtained between the adequate sample and inadequate sample, or between diagnostic success and diagnostic failure (*p* = 0.801 and *p* = 0.704, respectively). There was no difference in the sample adequacy and diagnostic success between radiologist A and radiologist B (*p* = 0.304 and *p* = 0.686, respectively).

### 3.3. Complications

Complication rates according to the biopsy target are shown in Table 2, and the list of complications is shown in Table 3. A total of 198 of the 1297 patients who underwent PNB had post-procedural complications (170 minor complications and 28 major complications). In the biopsy site for the lung and non-lung target, the occurrence of all complications (19.6%, 157/803 vs. 8.3%, 41/494, *p* < 0.001), minor complications (16.2%, 130/803 vs. 8.1%, 40/494, *p* < 0.001), and major complications (3.4%, 27/803 vs. 0.2%, 1/494, *p* < 0.001) were significantly different.

Out of the 28 major complications, 27 complications occurred during PNB of the lung target, while one biopsy resulted from PNB of the pleura. Pneumothorax occurred in 27 patients, which required chest tube placement, and hemothorax occurred in one patient, which required angioembolization. For the routine post-procedural chest radiograph performed during hospitalization, delayed pneumothorax was detected in four patients who underwent PNB for lung target.

### 3.4. Target Site Selection, Complications, and Diagnostic Yield According to PET/CT Timing

The biopsy site selection, post-procedural complication, and diagnostic yield according to PET/CT timing were investigated (Table 4). In the biopsy site selection, the decision for a non-lung target was significantly higher in the upfront PET/CT group than in the delayed PET/CT group for radiologist A (39.9%, 95/510 vs. 29.4, 128/787, *p* = 0.007), radiologist B (51.5%, 140/510 vs. 37.3%, 131/787, *p* = 0.001), and both radiologists (46.1%, 235/510 vs. 32.9%, 259/787, *p* < 0.001).

Major complications occurred significantly more frequently in patients of the delayed PET/CT group (2.9%; 23/787) compared to those in the upfront PET/CT group (1.0%, 5/510) (*p* = 0.031). The occurrence of all complications (14.3%, 73/510 vs. 15.9%, 125/787, *p* = 0.491) and minor complications (13.3%, 68/510 vs. 13.0%, 102/787, *p* = 0.912) did not differ significantly between the upfront PET/CT group and delayed PET/CT group. Sample adequacy (97.5% vs. 97.6%, *p* = 1.000) and diagnostic success rate (92.7% vs. 93.8%, *p* = 0.540) were not significantly different between the upfront PET/CT group and the delayed PET/CT group.

## 4. Discussion

This study demonstrated that the PNB for lung and non-lung targets in 1297 patients with newly diagnosed stage IV lung cancer was efficacious. Fewer minor (8.1% vs. 16.2%), major (0.2% vs. 3.4%), and overall complications (8.3% vs. 19.6%) occurred in patients who underwent PNB for non-lung target (*p* < 0.001 for all types of complications). In comparison to the delayed PET/CT group, the upfront PET/CT group exhibited a lower probability of lung target selection of PNB (53.9% vs. 67.1%, *p* < 0.001) and a reduced incidence of major complications (1.0% vs. 2.9%, *p* = 0.031), but did not significantly affect sample adequacy and diagnostic success rate.

Regarding PNB complications for lung target in patients with lung cancer, a meta-analysis revealed a total PNB complication rate of 17.3–38.8% and a very low rate of major complications (0.7–5.7%) [5]. PNB of the pleura, lymph nodes, and bone were associated with a low risk of minor complications (3.9–5.8%, 0–0.14%, and 0–2%, respectively), while no major complications were reported [4,10,11,12,14,27,28]. In our cohort of patients, the most commonly observed complication was pneumothorax, with complication rates similar to those reported in previous studies. The PNB for pleura had an incidence of 4.54% for pneumothorax, while PNB for the lung had a higher incidence of 6.66% [3,29]. Of the 27 chest tube insertions conducted for the major complications in our study, only one occurred in the pleura during PNB for the non-lung target, while all other complications were associated with PNB for the lung target. A case analysis was conducted regarding the major complications that occurred during PNB for the pleura and revealed that there was no pleural effusion, resulting in direct contact between the pleural lesion and the lung tissue. Parenchymal hemorrhage, hemoptysis, and hematoma were observed in patients who underwent PNB for the lung and non-lung target; however, these were self-limited.

PNB is a well-established procedure for the diagnosis of lung targets, with a reported diagnostic accuracy ranging from 85.7% to 95% [3,30]. Our results demonstrate a slightly higher diagnostic accuracy of 93.4%, which is within the above range. This is consistent with a previous study elucidating that the overall diagnostic accuracy of PNB is known to be higher for the diagnosis of malignancy compared to benign lesions [31]. The diagnostic accuracy of PNB for non-lung targets in patients with suspected lung cancer was as follows: lymph node (86–98%), bone (96–100%), pleura (85.4–88.4%), and liver (94.5–98.6%) [8,9,10,11,32,33]. Previous studies have performed PNB on various sites using diverse guidance modalities, which rendered it challenging to directly compare prior diagnostic accuracy. Similarly, for PNB performed on a non-lung target, our study shows a diagnostic accuracy of 93.4%.

A total of 494 patients with stage IV lung cancer underwent PNB in a non-lung target and 803 patients in a lung target performed by both radiologists, with a significant difference in the biopsy site selection between radiologist A and radiologist B (*p* < 0.001). It could be attributed to limited established guidelines for biopsy site decisions of PNB in patients with lung cancer, as well as the varying preferences of each radiologist and clinical physician at their hospitals. All radiologists showed an increased preference for a non-lung target for PNB when determining the biopsy site in the upfront PET/CT group compared to the delayed PET/CT group. PNB for lung target was significantly less frequent in the upfront PET/CT group compared to the delayed PET/CT group (275 vs. 528, *p* < 0.001). Most major complications, except for one hemothorax case arising from the pleura, occurred during PNB for lung target. The lower incidence of major complications in the upfront group may be attributed to fewer lung target compared to the delayed PET/CT group. PET/CT can identify a metabolically active lesion, which may lead to successful diagnoses [13,15,16]. The PNB based on PET/CT results demonstrated a high diagnostic success for lymph node (98.1%) or bone (96.1%) in lung cancer diagnosis [12,14]. Yoo et al. showed that PET/CT contributes to the practical determination of biopsy sites in patients with suspected malignancy, regardless of the determination of primary or metastatic lesions [13]. CT- and US-guided core needle biopsies have reported diagnostic accuracy ranging from 87.5% to 100% when assessing various sites in patients suspected of malignancy, including breast cancer, colorectal cancer, sarcoma, melanoma, prostate cancer, and lymphoma [34,35]. However, to date, few studies have been conducted on the impact of PET/CT in determining biopsy sites for patients with lung cancer. Our findings suggest that upfront PET/CT for determining the biopsy site of PNB is an effective approach in patients with lung cancer.

In our study, the procedure time of PNB was within the reasonable range for both lung target and non-lung target; however, the procedure time was significantly longer when targeting the lung, as compared to a non-lung target. US-guided PNB for the lung and pleura allows for a shorter procedure time, typically ranging from 321 to 2160 s, and can be performed during a single breath hold, while CT-guided PNB is longer with the procedure time ranging from 556 to 2700 s [4,32,36]. We presumed that performing PNB on a non-lung target reduced procedure time, attributed to the absence of the need for the patient’s respiratory control and differences in guidance modality selection. Specifically, we suggested that US-guided PNB was a commonly performed guidance modality for non-lung targets, which may contribute to the reduced procedure time of PNB on a non-lung target compared to PNB on a lung target. Based on the target site of PNB, no difference was found regarding hospital stay as the majority of patients who underwent PNB did not have any major complications that affected hospital stay.

An increased tissue sampling number of PNB for lung target, especially three or more specimens, was significantly associated with a higher diagnostic accuracy due to a reduced sampling error [37]. In contrast to their findings, our result showed that the number of cores obtained did not affect the sample adequacy and diagnostic accuracy of PNB in patients with stage IV lung cancer. Increasing the number of lymph node and pleura samplings during US-guided PNB failed to improve diagnostic accuracy, possibly due to the presence of necrotic tissue in the additional material obtained, which did not contribute to the diagnosis [10,27]. PNB was performed for various targets, not limited to the lung target. Thus, the number of cores obtained is not associated with sample adequacy and diagnostic accuracy in our study.

The current study has several limitations. First, it was a retrospective study with unintended selection bias. Second, chest radiography was used for the evaluation of delayed procedural complications, which may underestimate the delayed onset complication rate. However, regarding patient safety, any complications that were not detected on the chest radiography are considered negligible. In the diagnosis of lung cancer patients, the selection of the biopsy site could be influenced by various factors beyond PET/CT results, including the patient’s condition, clinical preferences of the physician, and the availability of hospital facilities and equipment. Lastly, our study exclusively focused on the investigation of the timing of PET/CT studies in stage IV lung cancer patients, indicating the necessity for conducting further research to evaluate the timing of PET/CT examinations in lung cancer patients.

In conclusion, PNB for non-lung targets in patients with stage IV lung cancer revealed significantly lower rates of complications compared to PNB for lung targets. The diagnostic yield of PNB according to the target site, including sample adequacy and diagnostic success rate, were not significantly different. Upfront PET/CT may have an impact on biopsy site selection for patients with advanced lung cancer, which could result in a lower rate of major complications with no change in the diagnostic yield. Larger-scale and multicenter prospective studies are needed to determine the appropriate timing for PET/CT examinations in lung cancer patients during clinical practice, facilitating easier decision-making for optimal biopsy site selection.

## Figures and Tables

**Figure 1 diagnostics-14-00153-f001:**
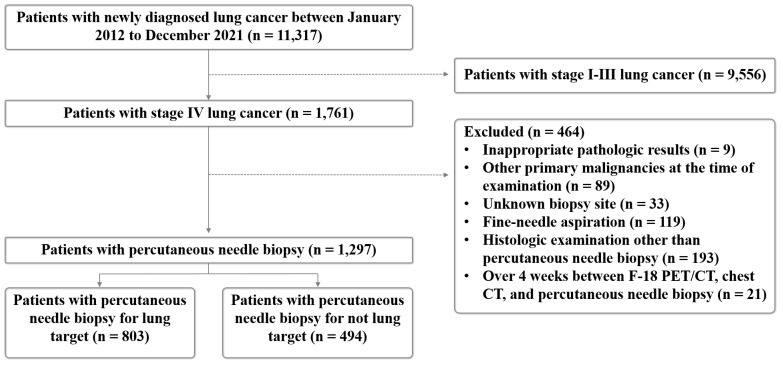
Flowchart for the inclusion and exclusion criteria for this study.

**Figure 2 diagnostics-14-00153-f002:**
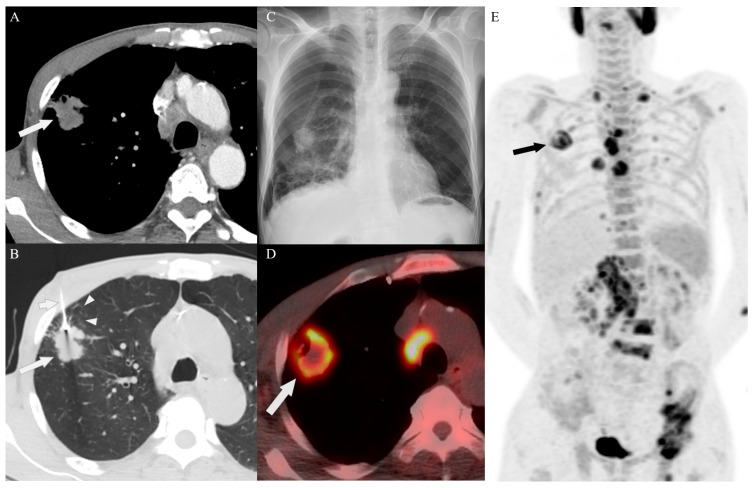
A 58-year-old man with sarcomatoid carcinoma in the delayed PET/CT group. (**A**) An enhanced axial CT image shows a heterogeneous enhancing nodule (arrow) in the upper right lobe. (**B**) An unenhanced axial CT scan shows the biopsy of the lesion (long arrow) in the supine position with the biopsy needle (short arrow). Note: Emphysema was detected along the biopsy needle passage adjacent to the nodule (arrowheads). (**C**) Pneumothorax was observed in the post-procedural chest radiography in the right hemithorax following percutaneous needle biopsy (PNB) of the lung. Subsequently, chest tube placement was performed to manage the pneumothorax. (**D**,**E**). After PNB for the lung target, fluorine-18 fluorodeoxyglucose (F-18 FDG) PET/CT imaging revealed hypermetabolic lesions in the lung (arrows), pleura, lymph nodes, right adrenal gland, and bones, including the left pelvis.

**Figure 3 diagnostics-14-00153-f003:**
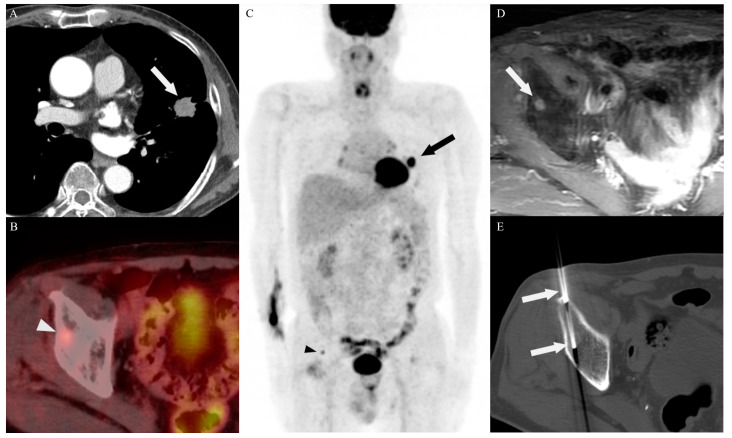
A 75-year-old man with adenocarcinoma in the upfront PET/CT group. (**A**) An enhanced axial CT image shows an irregular nodule (arrow) in the upper left lobe. (**B**,**C**) Prior to determining the target site of PNB, an F-18 FDG PET/CT scan revealed an increased uptake in the lung (arrow) and right acetabulum (arrowhead). (**D**) The increasing mass (arrow) observed on the axial gadolinium-enhanced fat-suppressed T1-weighted image in the right acetabular roof corresponds to the F-18 FDG uptake revealed in the same area as the previous PET/CT imaging. (**E**) CT-guided PNB for a metastatic lesion in the right acetabular roof was performed. An enhanced CT scan shows a 12-gauge bone biopsy needle (arrows) penetrating the metastatic lesion in the acetabulum. There were no complications observed following the procedure.

**Figure 4 diagnostics-14-00153-f004:**
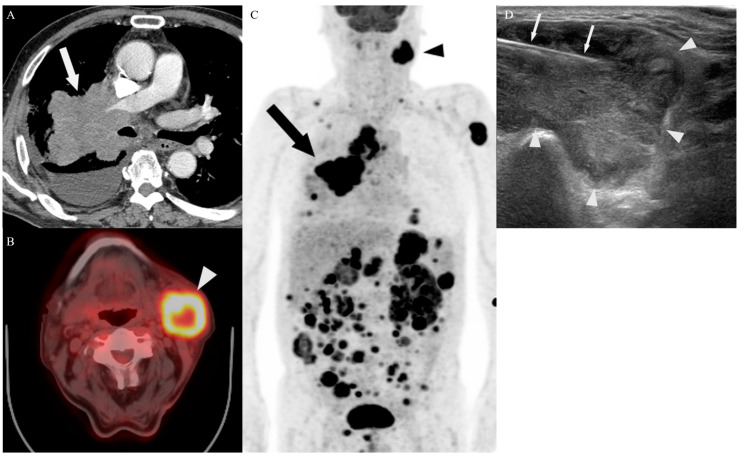
A 72-year-old man with small cell lung cancer in the upfront PET/CT group. (**A**) An enhanced axial CT image shows a large mass (arrow) in the upper right lobe, encasing the right pulmonary artery, with a right pleural effusion. (**B**,**C**) Prior to determining the target site of PNB, an F-18 FDG PET/CT scan shows an increased uptake in the lung (arrow), both adrenal glands, stomach, multiple soft tissues, lymph nodes including the mediastinum, left supraclavicular area, and left submandibular area (arrowhead). (**D**) US-guided PNB for the lymph node in the left submandibular area was performed. The biopsy needle (arrows) traversed the periphery of the lymph node (arrowheads). There were no complications observed following the procedure.

**Table 1 diagnostics-14-00153-t001:** Clinical characteristics of the study population.

Variables	Number of Patients	%
	(*n* = 1297)	
Sex		
Male	924	71.2
Female	373	28.8
Age † (in years)	71.4 ± 10.2	
PET/CT timing		
Upfront PET/CT group	510	39.3
Delayed PET/CT group	787	60.7
Biopsy technique		
Multiple passes	1261	97.2
Coaxial system	36	2.8
Location of lesions biopsied		
Lung	803	61.9
Non-lung	494	38.1
Lymph node	133	10.3
Pleura	205	15.8
Bone	88	6.8
Soft tissue	23	1.8
Kidney	2	0.2
Liver	43	3.3
Histological diagnosis		
Non-small cell lung cancer, not otherwise specified	181	14.0
Adenocarcinoma	692	53.4
Squamous cell carcinoma	237	18.3
Large cell carcinoma	13	1.0
Sarcomatoid carcinoma	10	0.8
Small cell lung cancer	164	12.6

† Continuous variables are presented as mean ± standard deviation.

**Table 2 diagnostics-14-00153-t002:** Characteristics, diagnostic yield, and complication rate of percutaneous needle biopsy.

	Non-Lung Target	Lung Target	Total	*p*-Value
	(*n* = 494)	(*n* = 803)	(*n* = 1297)	
Sex				0.856
Male	350 (70.9)	574 (71.5)	924 (71.2)	
Female	144 (29.1)	229 (28.5)	373 (28.8)	
Age † (in years)	71.4 ± 10.8	71.4 ± 9.9	71.4 ± 10.2	0.970
Guidance modality				<0.001 *
CT	31 (6.3)	346 (43.1)	377 (29.1)	
Cone-beam CT	0 (0)	117 (14.6)	117 (9.0)	
Ultrasound	463 (93.7)	340 (42.3)	803 (61.9)	
Number of cores obtained †	2.4 ± 1.0	1.6 ± 0.8	1.9 ± 1.0	<0.001 *
Procedure time, second †	643.7 ± 309.6	781.6 ± 374.3	729.1 ± 357.3	<0.001 *
Hospital stay, days †	9.8 ± 4.7	9.9 ± 7.5	9.9 ± 6.6	0.848
Variation in biopsy target selection				<0.001 *
Radiologist A	223 (45.1)	451 (56.2)	674 (52.0)	
Radiologist B	271 (54.9)	352 (43.8)	623 (48.0)	
Sample adequacy				0.629
Adequate	480 (97.2)	785 (97.8)	1265 (97.5)	
Inadequate	14 (2.8)	18 (2.2)	32 (2.5)	
Success rate				1.000
Diagnostic success	461 (93.3)	750 (93.4)	1211 (93.4)	
Diagnostic failure	33 (6.7)	53 (6.6)	86 (6.6)	
All complications	41 (8.3)	157 (19.6)	198 (15.3)	<0.001 *
Minor complications	40 (8.1)	130 (16.2)	170 (13.1)	<0.001 *
Major complications	1 (0.2)	27 (3.4)	28 (2.2)	<0.001 *

Note: Unless indicated otherwise, data are presented as a percentage in parentheses. * Statistically significant. † Continuous variables are presented as mean ± standard deviation.

**Table 3 diagnostics-14-00153-t003:** Major and minor complications.

Location of Lesions Biopsied	Results
Lung	
Minor complications	130 patients
Pneumothorax	83 cases
Hemoptysis	15 cases
Parenchymal hemorrhage	33 cases
Hematoma	4 cases
Hemothorax	1 case
Major complications	27 patients
Pneumothorax	26 cases
Hemothorax	1 case
Non-lung	
Lymph node	
Minor complications	25 patients
Hematoma	25 cases
Major complications	0 patient
Pleura	
Minor complications	5 patients
Pneumothorax	2 cases
Hemorrhage	2 cases
Hematoma	1 case
Major complications	1 patient
Pneumothorax	1 case
Bone	
Minor complications	10 patients
Hematoma	10 cases
Major complications	0 patient
Soft tissue, kidney, liver	
Minor and major complications	0 patient

**Table 4 diagnostics-14-00153-t004:** The selection of the target site, complications, and diagnostic yield according to the timing of PET/CT.

Variables	Upfront PET/CT Group	Delayed PET/CT Group	Total	*p*-Value
	(*n* = 510)	(*n* = 787)	(*n* = 1297)	
Target site performed by radiologists A and B				<0.001 *
Lung	275 (53.9)	528 (67.1)	803 (61.9)	
Non-lung	235 (46.1)	259 (32.9)	494 (38.1)	
Target site performed by radiologist A				0.007
Lung	143 (60.1)	308 (70.6)	451 (66.9)	
Non-lung	95 (39.9)	128 (29.4)	223 (33.1)	
Target site performed by radiologist B				0.001
Lung	132 (48.5)	220 (62.7)	352 (56.5)	
Non-lung	140 (51.5)	131 (37.3)	271 (43.5)	
All complications	73 (14.3)	125(15.9)	198 (15.3)	0.491
Minor complications	68 (13.3)	102 (13.0)	170 (13.1)	0.912
Major complications	5 (1.0)	23 (2.9)	28 (2.2)	0.031 *
Sample adequacy				1.000
Adequate	497 (97.5)	768 (97.6)	1265(97.5)	
Inadequate	13 (2.5)	19 (2.4)	32 (2.5)	
Success rate				0.540
Diagnostic success	473 (92.7)	738 (93.8)	1211(93.4)	
Diagnostic failure	37 (7.3)	49 (6.2)	86 (6.6)	

Note: Unless indicated otherwise, data are presented as a percentage in parentheses. * Statistically significant.

## Data Availability

The data are not publicly available due to ethical restrictions.

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
