# Peer review of "Clinical Role of Upfront F-18 FDG PET/CT in Determining Biopsy Sites for Lung Cancer Diagnosis"

_diagnostics, 2024, doi:10.3390/diagnostics14020153_

Round 1
Reviewer 1 Report
Comments and Suggestions for Authors
Thank you for submitting this interesting and informative manuscript to Diagnostics. I was pleased to receive it as a reviewer.
While your manuscript provides valuable insights into an important clinical topic, there are some areas that could be refined to further enhance the quality and impact of the work. Here are some respectful suggestions that could potentially improve the paper if you choose to implement them:
Title and Abstract
- The current title clearly conveys the main focus on PET/CT timing and biopsy outcomes. To broaden appeal, the authors could consider a title more generally encompassing the overall role of PET/CT in lung cancer diagnosis.
- In the abstract conclusion, the authors could reiterate the clinical need being addressed regarding optimizing lung cancer diagnosis, to emphasize real-world relevance.
Introduction
- The introduction establishes the clinical rationale and need for the study well. The authors could cite more guidelines on optimal PET/CT timing to strengthen the background. Describing potential debates around timing of PET/CT in guidelines could further set the stage for how the current study aids decision-making.
- The introduction nicely cites literature on risks of lung biopsy. Additionally citing studies on potential underdiagnosis or misdiagnosis without PET scanning could further highlight the need for optimal diagnostic approaches.
Discussion
- The authors interpret the study findings and implications adequately in the context of past literature. To add further perspective, comparison could be made with any similar studies involving PET-directed biopsy in other cancers rather than just lung cancer.
- Comparing the study biopsy diagnostic yield with that reported for primary lung cancers could provide meaningful context.
- The authors note that factors beyond PET/CT may influence site selection. Speculating on potential factors could offer additional perspectives.
Conclusions
- The conclusions summarize the main study results effectively. However, the authors could also discuss the need for future research to validate the study findings in larger, multicentre, prospective cohorts. This will emphasize that although promising, more corroboration is needed before changing clinical practice.
Overall, these suggestions aim to enhance the manuscript's quality and impact for clinicians and researchers. I believe that implementing some of the above suggestions would make your important work even stronger.
Author Response
Jan 08, 2024
Dear Reviewer:
Thank you for the opportunity to revise and resubmit our manuscript entitled, "Impact of Upfront F-18 FDG PET/CT Affecting Target Selection in Percutaneous Needle Biopsy for Lung Cancer Diagnosis: A Safety and Effectiveness Perspective" After carefully reading all points elaborated by the reviewers, we have revised our manuscript. Changed areas appear in gray opacity followed by the indication numbers of reviewers in red in an annotated version. Our specific responses are as follows:
RESPONSES TO THE COMMENTS OF REVIEWER:
Reviewer 1:
Thank you for submitting this interesting and informative manuscript to Diagnostics. I was pleased to receive it as a reviewer.
While your manuscript provides valuable insights into an important clinical topic, there are some areas that could be refined to further enhance the quality and impact of the work. Here are some respectful suggestions that could potentially improve the paper if you choose to implement them:
Title and Abstract
- The current title clearly conveys the main focus on PET/CT timing and biopsy outcomes. To broaden appeal, the authors could consider a title more generally encompassing the overall role of PET/CT in lung cancer diagnosis.
Response: Thank you for your comment. We fully agree with your opinion. Accordingly, we changed the title to “Clinical Role of Upfront F-18 FDG PET/CT in Determining Biopsy Sites for Lung Cancer Diagnosis”
- In the abstract conclusion, the authors could reiterate the clinical need being addressed regarding optimizing lung cancer diagnosis, to emphasize real-world relevance.
Response: Thank you for your comment. Accordingly, we added the sentence as follows: “Upfront PET/CT demonstrates potential clinical implications for enhancing the safety of lung cancer diagnosis in clinical practice.”
Introduction
- The introduction establishes the clinical rationale and need for the study well. The authors could cite more guidelines on optimal PET/CT timing to strengthen the background. Describing potential debates around timing of PET/CT in guidelines could further set the stage for how the current study aids decision-making.
Response: Thank you for your comment. We agree with your comment. We agreed revision was necessary for strengthen the background. The authors have included guidelines regarding optimal PET/CT timing for lung cancer diagnosis. However, these guidelines primarily focus on the reduction of additional procedure based on PET/CT timing. The authors, on the other hand, have emphasized the research background on PET/CT timing, with a focus on safety and diagnostic aspects. The paragraph was rewritten as follows.
“The use of fluorine-18 fluorodeoxyglucose (F-18 FDG) PET/CT has been recommended by several guidelines for lung cancer assessment. Detection of tumor viability through PET/CT improves the diagnostic yield of PNB for lung cancer diagnosis in patients who underwent biopsy of the lung or other structures. According to clinical guidelines by the National Comprehensive Cancer Network, the British Thoracic Society, and Cancer Council Australia the utilization of F-18 FDG PET/CT before additional diagnostic procedures is recommended. It has the capability to detect metastases, potentially offering an alternative means of obtaining tissue for pathological diagnosis. F-18 FDG-PET/CT serves a dual role by guiding the biopsy procedure and contributing to the assessment of disease stage. Performing F-18 PET/CT before biopsy in patients with lung cancer could potentially reduce the need for additional invasive procedures. When planning a diagnostic strategy for lung cancer, the diagnostic yield and potential risks involved should be carefully considered.
However, the correlation between the timing of F-18 FDG PET/CT in the diagnosis of lung cancer and its impact on the safety and diagnostic yield has been scarcely reported in the literature.”
- The introduction nicely cites literature on risks of lung biopsy. Additionally citing studies on potential underdiagnosis or misdiagnosis without PET scanning could further highlight the need for optimal diagnostic approaches.
Response: Thank you for your comment. There are studies suggesting that using PET/CT to guide needle placement in the viable portion of a lesion helps improve diagnostic accuracy. Accordingly, we added the sentence about improvement of diagnosis with PET/CT for lung cancer diagnosis in introduction section/
“Detection of tumor viability through PET/CT improves the diagnostic yield of PNB for lung cancer diagnosis in patients who underwent biopsy of the lung or other structures. Prior study comparing PNB with and without F-18 FDG PET/CT revealed that utilizing F-18 FDG PET/CT for guiding such biopsies enhances the accuracy of diagnosis. PNB guided by PET/CT findings exhibited a remarkable diagnostic success rate of 98.1% for lymph nodes and 96.1% for bone in the diagnosis of lung cancer.”
Discussion
- The authors interpret the study findings and implications adequately in the context of past literature. To add further perspective, comparison could be made with any similar studies involving PET-directed biopsy in other cancers rather than just lung cancer.
Response: Thank you for your comment. We could not find research that directly compared the diagnostic yield of PNB using PET-CT results for lung cancer diagnosis with diagnoses of other cancers. However, the diagnostic yield of PNB using PET-CT result has been reported for various types of cancers, with the majority demonstrating high yields. Accordingly, we added the sentence as follows:
“CT- and US-guided core needle biopsies have reported diagnostic accuracy ranging from 87.5% to 100% when assessing various sites in patients suspected of malignancy, including breast cancer, colorectal cancer, sarcoma, melanoma, prostate cancer, and lymphoma .”
- Comparing the study biopsy diagnostic yield with that reported for primary lung cancers could provide meaningful context.
Response: Thank you, we appreciate your comment. The authors fully agree with the opinions of the reviewers. During the initial manuscript drafting phase, we thoroughly reviewed the content. There are many studies available on the diagnostic yield of PNB for pulmonary lesions, demonstrating the established utility of PNB in this regard. However, most studies include cohorts encompassing both malignant and benign lung lesions, not limited to lung cancer alone. One study reported a yield of 48% for PNB in patients with lung cancer (1). Given the limited availability of direct comparisons with the current PNB diagnostic yield, attributed to the preliminary nature of the cited study, a detailed discussion on this aspect was omitted. The authors express willingness to incorporate and amend the manuscript if provided with literature evaluating PNB performance solely on cohorts comprising lung cancer patients not identified by the authors.
- Payne, C. R., Stovin, P. G., Barker, V., McVittie, S., & Stark, J. E. (1979). Diagnostic accuracy of cytology and biopsy in primary bronchial carcinoma. Thorax, 34(3), 294.
- The authors note that factors beyond PET/CT may influence site selection. Speculating on potential factors could offer additional perspectives.
Response: Thank you for your comment. Our clinicians gathered to engage in decision-making regarding the determination of biopsy sites at two medical institutions. For instance, in cases of suspected advanced lung cancer, where feasible biopsy sites exist both within and outside the lungs, factors such as procedural ease, patient preferences, and clinician inclinations could impact the selection of the biopsy site. Therefore, the authors have acknowledged these considerations in the limitations section and made some adjustments to the content.
“In the diagnosis of lung cancer patients, the selection of biopsy site could be influenced by various factors beyond PET/CT results, including the patient's condition, clinical preferences of the physician, and the availability of hospital facilities and equipment.”
Conclusions
- The conclusions summarize the main study results effectively. However, the authors could also discuss the need for future research to validate the study findings in larger, multicentre, prospective cohorts. This will emphasize that although promising, more corroboration is needed before changing clinical practice.
Response: Thank you for your comment. Accordingly, we added the sentence as follows in conclusion section:
“Larger-scale and multicenter prospective studies are needed to determine the appropriate timing for PET/CT examinations in lung cancer patients during clinical practice, facilitating easier decision-making for optimal biopsy site selection.”
- Overall, these suggestions aim to enhance the manuscript's quality and impact for clinicians and researchers. I believe that implementing some of the above suggestions would make your important work even stronger.
Response: I would like to express my sincere gratitude for dedicating your valuable time to review our paper. Your insightful advice emphasizing the clinical utility has been invaluable. We have revised the content accordingly, believing that it will be more beneficial clinically for our readers. Thank you sincerely for your guidance
With best wishes
Jongmin Park, MD

Reviewer 2 Report
Comments and Suggestions for Authors
The authors conducted a retrospective study investigating whether performing FDG PET before biopsy reduces needle biopsy-related complications. The study is clinically relevant, and the sample size was large. My comments are listed as follows,
1. The Figures demonstrated many cases, but need to follow the jounral guideline. However, the images for each case should be summarized into one figure. In addition, the figure legends showed images from A to E (or A to D in Figure 4), while those A-E were not demonstrated on the images.
2. Did the authors record the percentage of tumors in the specimen? In stage IV lung cancer, the specimen may undergo gene exams to test the presence of actionable mutations. Therefore, tumor abundance is essential. Did FDG PET/CT guided specimens have higher tumor abundance?
3. Table 1. Were the demographic data differ between upfront and delated PET/CT groups?
4. The reduced major complication rates in the upfront PET/CT group may be owing to fewer lung biopsies. The authors may explain this in the Discussion.
Author Response
Jan 08, 2024
Dear Reviewer:
Thank you for the opportunity to revise and resubmit our manuscript entitled, "Impact of Upfront F-18 FDG PET/CT Affecting Target Selection in Percutaneous Needle Biopsy for Lung Cancer Diagnosis: A Safety and Effectiveness Perspective" After carefully reading all points elaborated by the reviewers, we have revised our manuscript. Changed areas appear in gray opacity followed by the indication numbers of reviewers in red in an annotated version. Our specific responses are as follows:
RESPONSES TO THE COMMENTS OF REVIEWER:
Reviewer 2:
The authors conducted a retrospective study investigating whether performing FDG PET before biopsy reduces needle biopsy-related complications. The study is clinically relevant, and the sample size was large. My comments are listed as follows,
- The Figures demonstrated many cases, but need to follow the jounral guideline. However, the images for each case should be summarized into one figure. In addition, the figure legends showed images from A to E (or A to D in Figure 4), while those A-E were not demonstrated on the images.
Response: Thank you, we appreciate your comment. The authors referenced the "Preparing Figures, Schemes, and Tables" section on the website, along with the figure format of the recently published journal, to revise the figures in the manuscript.
Ref) Diagnostics 2024, 14(1), 116; Diagnostics 2024, 14(1), 112
- Did the authors record the percentage of tumors in the specimen? In stage IV lung cancer, the specimen may undergo gene exams to test the presence of actionable mutations. Therefore, tumor abundance is essential. Did FDG PET/CT guided specimens have higher tumor abundance?
Response: Thank you for your comment. This retrospective study has encountered limitations, particularly the lack of recorded tumor percentage within pathology reports from both hospital sites. Furthermore, the assessment of gene mutations was restricted to cases upon clinicians' request, thus not encompassed within the scope of this manuscript. We are sincerely thankful for the invaluable insights provided by the reviewers. Moving forward, our future explorations will involve conducting analyses on tumor abundance and gene mutations, employing subsequent reviews of pathology slides as part of our investigative approach.
- Table 1. Were the demographic data differ between upfront and delated PET/CT groups?
Response: Thank you for your comment. The demographic differences between the upfront and delayed PET/CT groups are as follows, as shown in the table below.
"Please see the new table in attachment file"
In the comparison between the upfront and delayed PET/CT groups, there were no statistically significant differences observed in age, sex, biopsy technique, or histological diagnosis. We added the sentence as follows in result section:
“No statistically significant differences were observed in age, sex, biopsy technique, and histological diagnosis between the upfront and delayed PET/CT groups.”
- The reduced major complication rates in the upfront PET/CT group may be owing to fewer lung biopsies. The authors may explain this in the Discussion.
Response: Thank you for your comment. We added the sentence as follows in discussion section
“PNB for lung target was significantly less frequent in the upfront PET/CT group compared to the delayed PET/CT group (275 vs. 528, p < 0.001). Most major complications, except for one hemothorax case arising from the pleura, occurred during PNB for lung target. The lower incidence of major complications in the upfront group may be attributed to fewer lung target compared to the delayed PET/CT group."
With best wishes
Jongmin Park, MD

Round 2
Reviewer 2 Report
Comments and Suggestions for Authors
The authors have addressed all my comments/suggestions.